# Therapeutic Effect of Renifolin F on Airway Allergy in an Ovalbumin-Induced Asthma Mouse Model In Vivo

**DOI:** 10.3390/molecules27123789

**Published:** 2022-06-12

**Authors:** Zhuya Yang, Xiaohong Li, Rongbing Fu, Min Hu, Yijie Wei, Xuhong Hu, Wenhong Tan, Xiaoyun Tong, Feng Huang

**Affiliations:** 1Key Laboratory of Yunnan Provincial Department of Education on Substance Benchmark Research of Ethnic Medicines, Yunnan University of Chinese Medicine, Kunming 650500, China; yangyzy321@163.com (Z.Y.); lixiaohong879@163.com (X.L.); 15087112997@163.com (R.F.); maomin199703@163.com (M.H.); weiyij2022@163.com (Y.W.); huxuhong4600@163.com (X.H.); twh85087@126.com (W.T.); 2The First Affiliated Hospital of Yunnan University of Chinese Medicine, Kunming 650021, China

**Keywords:** Renifolin F, *Shuteria involucrata* (Wall.) Wight & Arn., ILC2s, airway inflammation, allergic asthma, ovalbumin (OVA)

## Abstract

Renifolin F is a prenylated chalcone isolated from *Shuteria involucrata*, a traditional minority ethnic medicine used to treat the respiratory diseases and asthma. Based on the effects of the original medicine plant, we established an in vivo mouse model of allergic asthma using ovalbumin (OVA) as an inducer to evaluate the therapeutic effects of Renifolin F. In the research, mice were sensitized and challenged with OVA to establish an allergic asthma model to evaluate the effects of Renifolin F on allergic asthma. The airway hyper-reactivity (AHR) to methacholine, cytokine levels, ILC2s quantity and mircoRNA-155 expression were assessed. We discovered that Renifolin F attenuated AHR and airway inflammation in the OVA-induced asthmatic mouse model by inhibiting the regulation of ILC2s in the lung, thereby, reducing the upstream inflammatory cytokines IL-25, IL-33 and TSLP; the downstream inflammatory cytokines IL-4, IL-5, IL-9 and IL-13 of ILC2s; and the co-stimulatory factors IL-2 and IL-7; as well as the expression of microRNA-155 in the lung. The findings suggest a therapeutic potential of Renifolin F on OVA-induced airway inflammation.

## 1. Introduction

Allergic asthma is a chronic heterogeneous disease, affecting about 300 million people worldwide, and the burden of this disease is increasing in many countries [1]. It is characterized by eosinophilic inflammation, mucus hyper-production, airway hyper-responsiveness (AHR) and even airway remodeling [2].

The clinical manifestations of asthma are recurring wheezing, dyspnea, chest tightness and coughing, especially at night or in the early morning [3]. At present, the control of asthma is mainly using corticosteroids, β2-agonists and leukotriene receptor antagonists [4]. However, the effects of these therapies are not valid enough for 5–10% of severe asthma patients, and long-term use of these drugs can cause serious side effects [5]. Thus, it is crucial to find a safe anti-asthma treatments.

The majority of patients with allergic asthma suffer from a type 2 immune response, which is associated with Th2-mediated cytokines, including IL-4, IL-5, IL-9 and IL-13 [6]. However, recent studies have shown that mice lacking the adaptive immune system were still able to mount a type 2 response [7]. This indicates that there are other ways to develop allergic asthma.

Innate lymphoid cells (ILCs) are recognized as a key component of the innate immune system to protect the human body against many diseases [8]. ILC2s are a subset of ILCs that exhibit lymphoid morphology but lack lymphocyte lineage markers [9]. ILC2s are closely associated with the pathogenesis of asthma [10]. Exposure to allergens could lead to airway damage and induce the production of alarmin, such as IL-25, IL-33 and thymic stromal lymphopoietin (TSLP) [11,12]. These cytokines can directly induce the production of type 2 cytokine in ILC2s, leading to type 2 inflammation [13]. Therefore, inhibition of ILC2s mediators and activators could be a therapeutic option for the allergic asthma.

Medicinal plants are plants that have therapeutic effects on diseases. Compounds isolated from some medicinal plants showed potential efficacy in anti-inflammation, or even anti-asthma [14,15,16]. Renifolin F (Figure 1) is a prenylated chalcone that was first isolated from *Desmodium renifolium*, and the effect of this ingredient in vivo has not been reported yet [17]. In the research, Renifolin F was isolated from another medicinal plant *Shuteria involucrata* (Wall.) Wight & Arn., which is called “Tong-qian-ma-huang” in China and has been used as a traditional *Dai* ethnic medicine for the treatment of the inflammation of bronchi and lungs and also be used to treat asthma [18,19]. As one of the main components of this traditional *Dai* ethnic medicine, here, for the first time, the paper reports the anti-allergic airway effect of Renifolin F in an ovalbumin-induced asthma mouse model in vivo.

## 2. Results

### 2.1. Effects of Renifolin F on Total Serum IgE and Allergen-Specific IgE Production in Mice Serum

Since IgE is one of the most important factors related to the atopic disease, including allergic asthma [20], we investigated the levels of total IgE and allergen-specific IgE in mice serum. Compared with the normal control, OVA significantly enhanced the production of total IgE and allergen-specific IgE in serum of mice (*p* < 0.01). Both Dex (1.0 mg/kg) and Renifolin F (3.0 and 1.5 mg/kg) could notably reduce the production of total IgE and allergen-specific IgE in OVA-induced mice (*p* < 0.01). Notably, the inhibitory effect of Renifolin F on the production of total IgE was dose-dependent (Figure 2A).

### 2.2. Effects of Renifolin F on Eotaxin, IL-4, IL-5, IL-13, IL-9 and IL-6 Levels in Mice BALF

Eotaxin is a small protein that can stimulate the migration of eosinophils from the small blood vessels in the lungs [21]. Th2 cytokines are highly expressed in allergic inflammation [6]. Therefore, eotaxin and Th2 cytokines, including IL-4, IL-5, IL-13, IL-9 and IL-6, were the key factors of type 2 immune response. We measured the production of these important cytokines using ELISA. The cytokines were significantly increased in the OVA-treated group (Model) compared to the normal control group (NC) (Figure 2B). However, treatment of Renifolin F (3.0 and 1.5 mg/kg) reduced the levels of these cytokines in the BALF of OVA-induced asthma mice, and the Dex group showed similar effects. Among these cytokines, the levels of IL-5 and IL-9 were reduced in a dose-dependent manner.

### 2.3. Effects of Renifolin F on Symptoms and Muc5ac in OVA-Induced Allergic Asthma Mice

To assess the anti-inflammatory effects of Renifolin F, we performed histological analysis of the lung tissues using H&E and PAS staining. The lungs of the Model group mice showed dense perivascular and peribronchial infiltration of inflammatory cells around the trachea and alveoli, the severely hyperplastic goblet cells on the airway epithelium, distinct thickening of the basement membrane and mucus secretion within the bronchi when compared to the morphology of normal control tissues. Renifolin F (3.0 and 1.5 mg/kg) suppressed the increased infiltration of inflammatory cells and the mucus secretion (Figure 3A). The inflammation and mucus score revealed that Renifolin F had an inhibitory effect against airway inflammation (Figure 3B).

The effect of Renifolin F on Airway Hyperresponsiveness (AHR), one of the characteristics of asthma, was explored by measuring the airway function in the mice. The inspiratory resistance (RI) of each group was measured through the BUXCO. The RI in the OVA-induced group was significantly higher than that in normal control at all doses of MCh, thus, indicating that the gas exchange function of the lung was hindered after OVA treatment (Figure 4). These levels were significantly attenuated by RF-H (3.0 mg/kg) treatment at 6.25–12.5 mg/mL of MCh. The findings indicate that administering Renifolin F effectively reduced AHR in the asthma mice model.

Another important feature of asthma is mucus hyper-production. Muc5ac is one of the major gel-forming mucins in goblet cells, which are the major secretory cells involved in airway epithelium [22]. Hence, we measured the level of Muc5ac in mice (Figure 5). Compared with Normal control, OVA significantly enhanced the production of Muc5ac in BALF of the mice (*p* < 0.01). Both Dex and Renifolin F reduced the level of Muc5ac in OVA-induced mice in a dose-dependent manner (*p* < 0.01).

### 2.4. Effects of Renifolin F on ILC2s Cells and Its Upstream Cytokines in the Lung

We evaluated the levels of IL-33, IL-25 and TSLP, which are the upstream cytokines of ILC2s, in mice BALF when the airway epithelium was exposed to allergens to confirm whether Renifolin F inhibited OVA-induced allergic asthma by acting on ILC2s in mice. IL-2 and IL-7 promote the development of ILC2s; therefore, we also evaluated the levels of IL-2 and IL-7 in mice BALF. We found that Renifolin F treatment reduced the levels of IL-25, IL-33, TSLP, IL-2 and IL-7 in mice BALF, and the levels of TSLP, IL-2 and IL-7 were reduced in a dose-dependent manner (Figure 6).

To explore the effect of Renifolin F on ILC2s expansion, the proportion of ILC2s in the lung of the mice was determined by flow cytometry. ILC2s were negative for lineage markers (CD3ε, NK1.1, Ly-6G and Ly-6C (Gr-1), F4/80, CD5, CD8α, CD11c, CD19, FceRIα and TER119) and positive for CD45, CD127 and ST2. A significant increase of ILC2 population was found in the Model group compared with that of the Normal control group (*p* < 0.01) (Figure 7). After Renifolin F (1.5 and 3.0 mg/kg) treatment, the number of ILC2s in the lung showed a striking reduction (*p* < 0.01).

### 2.5. Effects of Renifolin F on the Expression of microRNA-155 in the Lung of Allergic Asthma Mice

MicroRNA-155 is associated with the production of a variety of inflammatory cells and is an important gene that regulates airway inflammation. Among the cells that are regulated by miroRNA-155, ILC2s is an important inflammatory cell in airway inflammation. Studies have shown that ILC2s are regulated by microRNA-155 in allergic airway inflammation [23]. To explore the effect of Renifolin F on ILC2s, we measured the expression of microRNA-155 in allergic asthma mice. We found that Renifolin F (3.0 and 1.5 mg/kg) treatment reduced the expression level of microRNA-155 in OVA-induced mice in a dose-dependent manner (Figure 8).

## 3. Discussion

Asthma is a common heterogeneous respiratory system disease characterized by chronic airway inflammation, airway hyperresponsiveness, mucus hyper-production and airway remodeling. It includes many subtypes with a large number of patients caused by allergens [24]. At present, the treatment of asthma is mainly based on medium-dose inhaled corticosteroids (ICS) or low-dose ICS combined with long-acting β2-agonists (LABA). These treatment methods are helpful for some patients who suffer from asthma. However, the long-term use of these medicines will bring greater side effects to the development of patients, especially children, and the symptoms of some asthma patients cannot be controlled solely by those medicines [25]. Therefore, there is an urgent need to find a new drug with high efficiency and low side effects for the treatment of allergic asthma.

Studies have shown that BALB/c mice can be sensitized by ovalbumin to induce an allergic asthma model [26]. OVA-induced asthma models can mimic allergic states. In the allergic asthma mice, high levels of IgE antibodies were produced in the peripheral blood and clear Th2 responses were observed. Type 2 immune responses typically lead to increased eosinophils and clusters in the inflammation location, which is one of the characteristics of type 2 immunopathology [27,28]. In our study, compared with the normal control group, the lung tissues of the model group had distinct inflammatory cell infiltration and airway goblet cell proliferation accompanied with increased levels of total IgE, allergen-specific IgE and Muc5ac. 

This suggests that ovalbumin successfully induced the eosinophilic asthma-like characteristics of BALB/c mice, which manifested as airway inflammation, airway hyperresponsiveness and airway mucus secretion. Dex was used as a positive control drug in our study. The results showed that similar to Dex, Renifolin F (3.0 and 1.5 mg/kg) can significantly improve the pathological features of asthma and reduce the occurrence of airway hyperresponsiveness in OVA-induced allergic asthma mice, and the levels of mucin and related inflammatory factors in mice serum and BALF were inhibited.

At the onset of allergic asthma, the Th2 immune response is always dominant. Th2 cells secrete IL-4, IL-5, IL-9, IL-13 and other inflammatory factors, among which IL-4 and IL-13 are the regulators of asthma. These cytokines are the key to airway inflammation. They stimulate B cells to produce IgE and mucus, remodel epithelial cells, activate M2 macrophages and bind to the IgE FcεRI receptors on the surface of mast cells and basophils, thereby, causing the release of histamine and leukotrienes. IL-5 plays a key role in the differentiation, maturation, chemotaxis and activation of eosinophils and often cooperates with IL-4 to upregulate the level of eosinophils in patients with allergic asthma [29,30,31,32,33]. 

Therefore, Renifolin F may reduce the levels of total IgE and allergen-specific IgE in serum by inhibiting IL-4 and IL-5. As the main product of Th9 cells, IL-9 is a pleiotropic cytokine and can stimulate the proliferation of activated T cells, promote the proliferation and differentiation of mast cells, increase the production of IgE by B cells, upregulate high-affinity IgE receptors and induce the production of IL-6 [34,35]. Activated IL-13 and IL-9 can promote allergen-induced airway hyperresponsiveness [36,37,38]. The ability of Renifolin F to inhibit AHR may be related to its ability to inhibit the production of IL-13 and IL-9.

In addition to Th2 cells, ILC2s are another source of type 2 cytokines. In recent years, the innate immune cells (ILCs) have received great attention for their roles in inflammatory diseases and as defense cells against pathogens and foreign antigens. Although ILCs have lymphoid morphology, unlike T cells and B cells, they do not have the microstructure of an antigen receptor to recognize antigens. According to the expression of cytokines and transcription factors, ILCs are divided into three different subtypes: group 1 ILCs (ILC1s), group 2 ILCs (ILC2s) and group 3 ILCs (ILC3s). ILC2s, like Th2 cells, can induce a type 2 immune response and produce type 2 cytokines, and thus they are important participants in the type 2 immune response and allergic asthma [8,9,39].

The airway is an organ that communicates with the external environment and is constantly exposed to pathogens and foreign antigens in the air. Therefore, it is the first step of host defense [40]. The airway epithelium produces IL-25, IL-33 and TSLP after being stimulated by allergens, such as ovalbumin, house dust mites, pollen and other substances [41]. Studies have shown that IL-25 derived from epithelial cells can activate ILC2s and promote the production of type 2 cytokines in OVA-induced asthma models, suggesting that IL-25 may play a key role in the development of asthma [42]. IL-33 is constantly expressed by mucosal barrier cells, released from the cells as an “alarm protein” and repairs the damaged site. 

IL-33 is considered to be one of the main activators of ILC2s that induces the production of type 2 cytokines [43]. Similar to the first two epithelial-derived cytokines, TSLP is also a major activator of ILC2s, which can induce the production of type 2 cytokines, and the continuous production and release of TSLP in the airway may lead to the occurrence of moderate to severe asthma that is resistant to steroids [44]. The above three epithelial-derived cytokines can induce ILC2s to secrete Th2-type cytokines, such as IL-4, IL-5, IL-9 and IL-13 [45]. IL-2 and IL-7 promote the induction of IL-4, IL-5 and IL-13 by activating the STAT5 signaling pathway in ILC2s and mediate the type 2 inflammatory response [30].

According to the results of our study, Renifolin F not only can reduce the upstream cytokines of ILC2s (such as IL-25, IL-33 and TSLP) in a dose-dependent manner, thereby reducing the production of ILC2s cells but can also reduce the downstream cytokines of ILC2s (such as IL-4, IL-5, IL-9 and IL-13) in a dose-dependent manner, subsequently reducing the content of IL-2 and IL-7, which are co-stimulatory factors that activate ILC2s. These results indicate that the anti-asthma effect of Renifolin F may associate with its ability to inhibit the production of ILC2s in lung tissues.

At the same time, we found that Renifolin F can downregulate the expression of microRNA-155, an important regulator of ILC2s. MicroRNA (miRNA) is a conserved single-stranded RNA molecule (approximately 22 nucleotides in length) that regulates gene expression by targeting the 3’UTR of the mRNA transcript [46]. MiRNAs act to inhibit translation and destabilize the target mRNA, resulting in reduced protein production. Therefore, miRNA expression plays an important role in regulating the differentiation and functions of immune cells, such as apoptosis, cancer, development and inflammation [47]. 

Notably, chronic airway inflammation is characteristic of asthma. It has been proven that multiple miRNAs are related to the pathogenesis of allergic asthma. Specifically, microRNA-155 plays an important role in regulating allergen-induced acute and chronic airway inflammation [48]. MicroRNA-155 is located in the B cell integration cluster (BIC) gene, which is necessary for B cells to produce antigen-specific IgG1 antibodies. In response to Toll-like receptor signals, microRNA-155 can downregulate the expression of the FcεRI receptor itself via the transcription factor PU, through which it regulates the functions of mast cells and dendritic cells [23].

The toll-like receptor (TLR) is one of the most studied pathogen detection systems and plays a central role in host defense [49]. In a type 2 immune response, IL-33 binds to the receptors ST2 and IL-1RAcP to form a heterodimer. MyD88 is then activated as a key linker molecule in the TLR signaling pathway. Afterword, IRAK1 and IRAK4 are recruited into the receptor complex to induce the activation of downstream signaling molecules NFκB, IκBα, ERK, MAPK, JNK1, etc., which then activate ILC2s and produce type 2 cytokines (IL-4, IL-5, IL-9, IL-13, etc.) to mediate allergic airway inflammation [50].

Therefore, ILC2 relies on microRNA-155 to function. The expression of microRNA-155 increases in tissues with ILC2s cell-mediated inflammation, which was activated by IL-33 and accompanied by increased IL-13 production [51]. Our results showed that Renifolin F could downregulate the expression of microRNA-155, possibly by reducing the production of IL-33 to inhibit the activation of the IL-33/ST2 axis, thereby, reducing the production of ILC2s to prevent the occurrence of airway inflammation.

Renifolin F is a natural product that is isolated from *Shuteria involucrata*, which has been used as a traditional Dai medicine named “Tong-qian-ma-huang”. “Tong-qian-ma-huang” has the effect of relieving cough and asthma. However, there is no report on the therapeutic efficacy of Renifolin F. Therefore, in this study, we performed an anti-allergic asthma pharmacodynamic analysis of this compound. We first demonstrated that Renifolin F attenuated OVA-induced allergic asthma, which may be achieved by inhibiting the production of ILC2s, suppressing the expression of its key regulator microRNA-155 and consequently inhibiting upstream and downstream inflammatory cytokines. This study proposes a potential new medicine for the treatment of allergic asthma.

## 4. Materials and Methods

### 4.1. Plant Materials and Renifolin F Analysis

The roots of *Shuteria involucrata* were collected in June 2019 in Pu’er of Yunnan province, People’s Republic of China. Renifolin F was isolated from the dried and powdered roots of *S. involucrata* (10.0 kg) with 70% EtOH at room temperature and concentrated in vacuo to yield a crud extract. The EtOH crude extract was diluted with water and extracted with ethyl acetate. The EtOAc fraction was purified by MCI-gel and Sephadex LH-20 and eluted with a gradient of H_2_O-MeOH (1:0→0:1) and silica gel (CHCl_3_-MeOH, 20:1→5:1). Finally, 117 mg Renifolin F was obtained. Purified Renifolin F compound was verified by comparing ^1^H and ^13^C-NMR, 2D NMR and MS data with literature values (Li et al., 2014) (Appendix A). Based on HPLC analysis, the purity of Renifolin F was higher than 98.0% (Appendix A).

### 4.2. Mice

Pathogen-free female BALB/c mice (4–6 weeks old; weighing 18–22 g) were purchased from Liaoning Changsheng Biotechnology Co., Ltd. (license number SCXK 2015-0001, Liaoning, China). The animals were housed at room temperature (20–25 °C) with constant humidity (40–70%) under a 12/12 h (light/dark) cycle in a pathogen-free rodent facility for one week before treatment. Food and water were provided to the mice ad libitum. The animal ethics committee of Yunnan University of Chinese Medicine approved the animal experimental procedures and welfare (No. R-06202032).

### 4.3. OVA-Induced Allergic Asthma Model and Treatment

BALB/c mice were sensitized on days 1, 8 and 15 via an intraperitoneal injection of 0.2 mg of OVA (Sigma-Aldrich, St. Louis, MO, USA) emulsified in 1 mg of aluminum hydroxide in a total volume of 0.2 mL. The Normal control mice only received an intraperitoneal injection of 0.2 mL saline. On days 21–27 after the initial sensitization, the mice were challenged for 30 min once a day with an aerosol of 2% (weight/volume) OVA in saline (or saline alone for the Normal control) using an ultrasonic nebulizer.

After the induction process, the experimental animals were divided into five groups, with six mice in each group: (1) Normal control (saline-induced, i.g.); (2) Model (OVA-induced, i.g.); (3) Dex (OVA-induced, treated with Dex (Sigma-Aldrich, St. Louis, MO, USA) at 1.0 mg/kg, i.g.); (4) RF-H (OVA-induced, treated with Renifolin F at 3.0 mg/kg, i.g.); and (5) RF-L (OVA-induced, treated with Renifolin F at 1.5 mg/kg, i.g.). The treatments were given to the mice from the 21st day to the 27th day, once a day 1 h before each OVA challenge. The mice were sacrificed 24 h after the last OVA aerosol challenge for different analyses.

### 4.4. Collection and Management of Serum

The peripheral blood of mice was collected, stood at room temperature for 1 h and centrifuged at 800× *g* for 10 min at 4 °C. The supernatant was prepared and stored at −80 °C for the analysis of total serum IgE and allergen-specific IgE.

### 4.5. Collection and Management of the Mice Bronchoalveloar Lavage Fluid (BALF)

The mice were sacrificed, and the trachea of each mouse was exposed. A tracheal cannula was inserted into the trachea. The airways of left lungs were lavaged twice with 0.3 mL PBS to obtain BALF. The BALF was centrifuged at 800× *g* for 10 min at 4 °C, and the supernatant was collected for the determination of cytokine production.

### 4.6. Measurement of Cytokine Production Levels by ELISA

The concentrations of Eotaxin, IL-2, IL-4, IL-5, IL-6, IL-7, IL-9, IL-13, IL-25, IL-33, TSLP, total serum IgE (MultiSciences Biotech Co., Ltd., Hangzhou, China), Muc5ac (Elabscience Biotechnology Co., Ltd, Wuhan, China) and allergen-specific IgE (Cayman Chemical, Ann Arbor, MI, USA) were measured by ELISA, according to the manufacturer’s instructions.

### 4.7. Assessment of Airway Hyperresponsiveness (AHR)

The effect of Renifolin F on AHR was measured within 24 h following the last allergen challenge. After tracheotomy, anesthetized mice were connected to the Resistance and Compliance System (RC system, BUXCO, Wilmington, NC, USA) and challenged with increasing doses of MCh (0, 3.125, 6.25, 12.5, 25 and 50 mg/mL). The inspiratory resistance (RI) was measured at every dose to calculate the percentage of each concentration to the baseline. 

### 4.8. Histological Analysis of Lung Tissue

The left lung was lavaged with PBS, and the middle part of right lung tissue was fixed in 4% (*v*/*v*) formaldehyde for 48 h. Afterward, the sample tissue was dehydrated in various concentrations of ethanol, embedded in paraffin and cut with a microtome at a thickness of 4 μm/section. The sections were deparaffinized and stained with hematoxylin and eosin (H&E, Sigma-Aldrich Inc., Steinheim am Albuch, Germany), followed by the periodic acid Schiff reagent (PAS, Wuhan Servicebio Technology Co., Ltd., Wuhan, China). The lung tissue slices were evaluated under a light microscope to observe the infiltration of inflammatory cells and the airway morphology.

### 4.9. Flow Cytometric Analysis

Lung tissue was digested in 50 μg/mL Liberase TM (1:100) + 1 μg/mL DNase I (1:200). A cell suspension was collected. To detect ILC2s, antibodies for surface antigens were incubated for 30 min, resuspended in FACS buffer and separated on a LSR-Fortessa (BD Pharmingen, San Diego, CA, USA) based on the surface-marker expression. The following antibodies were used: lineage-APC Streptavidin (BD Pharmingen) markers (CD3ε (BD Pharmingen), CD4 (eBioscience, San Diego, CA, USA), CD8a (BD Pharmingen), CD11c (eBioscience), FceRIa (BioLegend), NK1.1 (BioLegend), CD19 (eBioscience), TER119 (eBioscience), CD5 (BD Pharmingen), F4/80 (eBioscience), Ly-6G and Ly-6C (Gr-1, BD Pharmingen)), anti-7AAD (BD Pharmingen), anti-CD45-APCcy7 (BD Pharmingen), anti-ST2-PE (BD Pharmingen) and anti-CD127-FITC (BD Pharmingen).

### 4.10. RNA Isolation and microRNA-155 Expression

The total RNA was extracted from the lung tissue using Trizol Reagent (Servicebio, Wuhan, China), according to the manufacturer’s protocol. The expression of microRNA-155 was measured using mmu-miR-155-5p primer. The sequences of the primers were as follows: GTCGTATCCAGTGCAGGGTCCGAGGTATTCGCACTGGATACGACACCCCTA (forward) and CCTCGTTAATGCTAATTGTGA (reverse).

### 4.11. Statistical Analysis

The data were analyzed and graphed with GraphPad Prism 6.0 (GraphPad, San Diego, CA, USA) and expressed as the mean ± SEM. Statistical analysis was performed using analysis of variance (ANOVA), followed by a multiple comparison test with Bonferroni adjustment. A value of *p* < 0.05 was considered significant.

## 5. Conclusions

We demonstrated that Renifolin F attenuated allergic inflammation in an OVA-induced allergic asthma mice model in vivo, and the activity of Renifolin F was assessed through inhibiting the regulation of ILC2s in the lung, including inhibition of the upstream cytokines of ILC2s (such as IL-25, IL-33 and TSLP) and the downstream cytokines of ILC2s (such as IL-4, IL-5, IL-9 and IL-13), subsequently reducing the content of IL-2 and IL-7, which are co-stimulatory factors that activate ILC2s. At the same time, there was a reduction in the expression of microRNA-155, one of the regulation genes of ILC2s. These results indicate that the anti-asthma effect of Renifolin F may be associated with its ability to inhibit the production of ILC2s in lung tissues. These findings will be useful for the development of Renifolin F as a novel anti-asthmatic drug. 

## Figures and Tables

**Figure 1 molecules-27-03789-f001:**
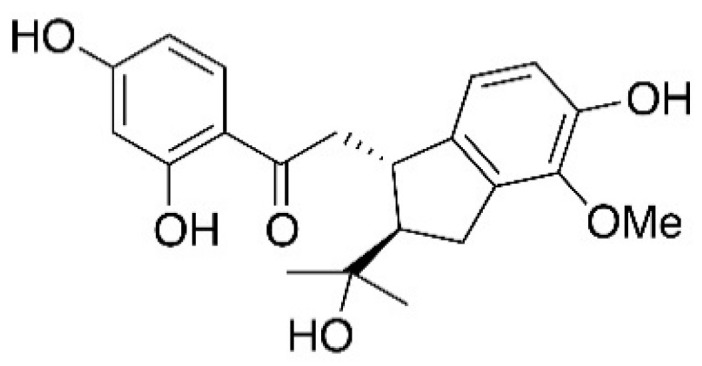
Chemical structure of Renifolin F.

**Figure 2 molecules-27-03789-f002:**
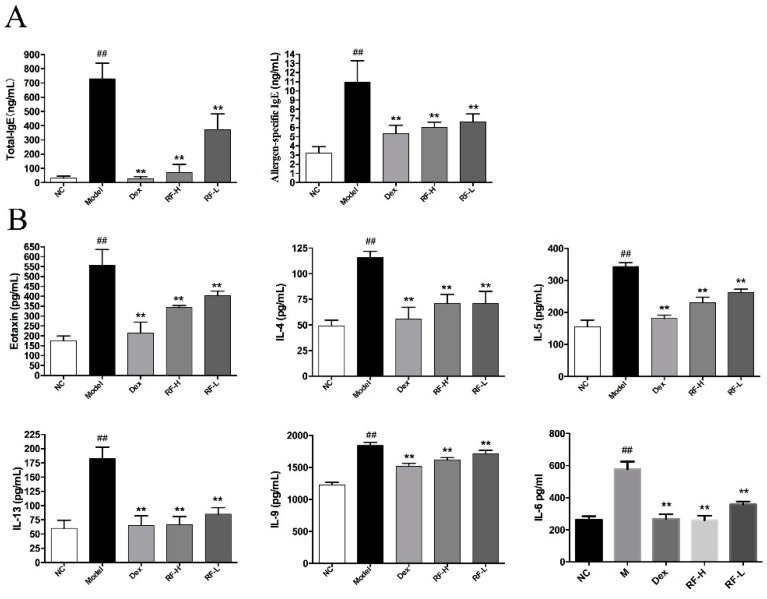
The effects of Renifolin F on cytokine production in serum and BALF. (**A**) The levels of total IgE and allergen-specific IgE in serum. (**B**) The levels of IL-4, IL-5, IL-13, IL-9 and IL-6 in BALF. Normal control, saline-induced mice only. Model, OVA-induced mice. Dex, OVA-induced mice treated with Dex (1.0 mg/kg). RF-H, OVA-induced mice treated with Renifolin F (3.0 mg/kg). RF-L, OVA-induced mice treated with Renifolin F (1.5 mg/kg). Statistics represented as the mean ± SEM (*n* = 6). ## *p* < 0.01 vs. Normal control and ** *p* < 0.01 vs. Model.

**Figure 3 molecules-27-03789-f003:**
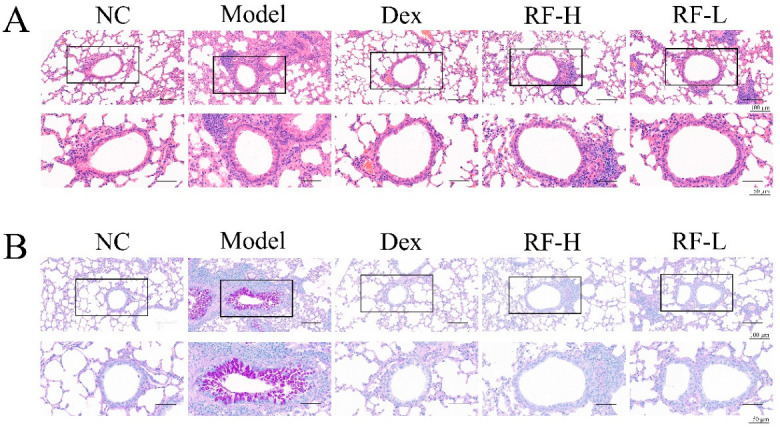
The effects of Renifolin F on histopathologic changes in the lung tissues of OVA-induced allergic mice in vivo. (**A**) Lung tissue sections were stained with H&E. (**B**) Lung tissue sections were stained with PAS. Normal control, saline-induced mice only. Model, OVA-induced mice. Dex, OVA-induced mice treated with Dex (1.0 mg/kg). RF-H, OVA-induced mice treated with Renifolin F (3.0 mg/kg). RF-L, OVA-induced mice treated with Renifolin F (1.5 mg/kg).

**Figure 4 molecules-27-03789-f004:**
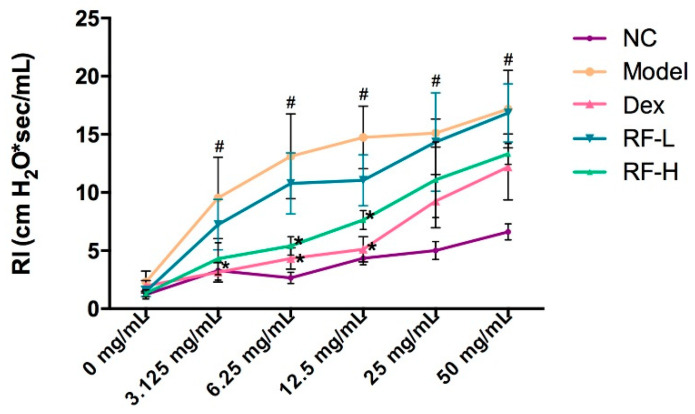
At 24 h after the last challenge, the inspiratory resistance (RI) was measured at the challenge of MCh. Normal control, saline-induced mice only. Model, OVA-induced mice. Dex, OVA-induced mice treated with Dex (1.0 mg/kg). RF-H, OVA-induced mice treated with Renifolin F (3.0 mg/kg). RF-L, OVA-induced mice treated with Renifolin F (1.5 mg/kg). Statistics represented as the mean ± SEM (*n* = 6). # *p* < 0.05 vs. Normal control, and * *p* < 0.05 vs. Model.

**Figure 5 molecules-27-03789-f005:**
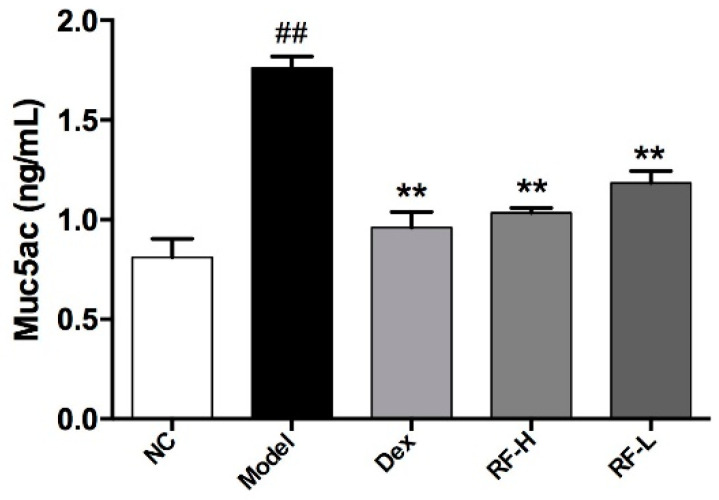
The effects of Renifolin F on Muc5ac levels in BALF. Normal control, saline-induced mice only. Model, OVA-induced mice. Dex, OVA-induced mice treated with Dex (1.0 mg/kg). RF-H, OVA-induced mice treated with Renifolin F (3.0 mg/kg). RF-L, OVA-induced mice treated with Renifolin F (1.5 mg/kg). Statistics represented as the mean ± SEM (*n* = 6). ## *p* < 0.01 vs. Normal control, and ** *p* < 0.01 vs. Model.

**Figure 6 molecules-27-03789-f006:**
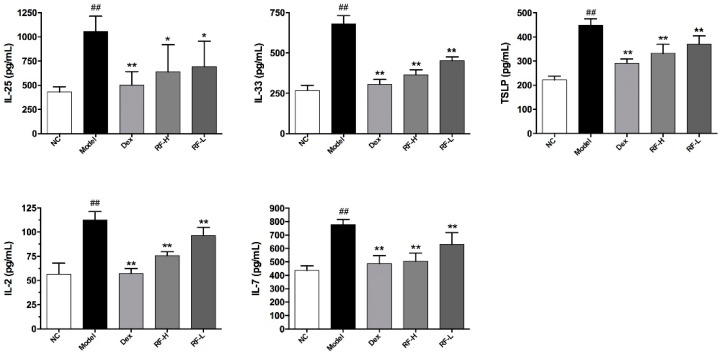
The effects of Renifolin F on IL-25, IL-33, TSLP, IL-2 and IL-7 levels in BALF. Normal control, saline-induced mice only. Model, OVA-induced mice. Dex, OVA-induced mice treated with Dex (1.0 mg/kg). RF-H, OVA-induced mice treated with Renifolin F (3.0 mg/kg). RF-L, OVA-induced mice treated with Renifolin F (1.5 mg/kg). Statistics represented as the mean ± SEM (*n* = 6). ## *p* < 0.01 vs. Normal control, * *p* < 0.05 vs. Model and ** *p* < 0.01 vs. Model.

**Figure 7 molecules-27-03789-f007:**
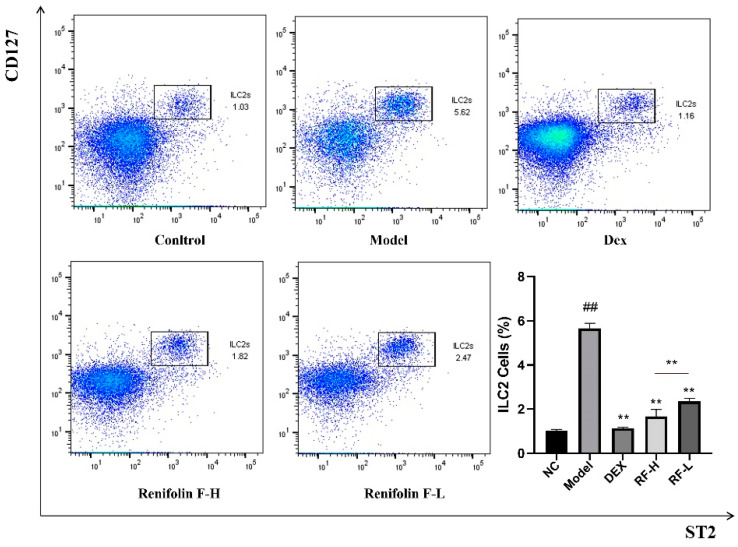
The effects of Renifolin F on the frequency of ILC2s cells in the lung. Normal control, saline-induced mice only. Model, OVA-induced mice. Dex, OVA-induced mice treated with Dex (1.0 mg/kg). RF-H, OVA-induced mice treated with Renifolin F (3.0 mg/kg). RF-L, OVA-induced mice treated with Renifolin F (1.5 mg/kg). Statistics represented as the mean ± SEM (*n* = 6). ## *p* < 0.01 vs. Normal control and ** *p* < 0.01 vs. Model.

**Figure 8 molecules-27-03789-f008:**
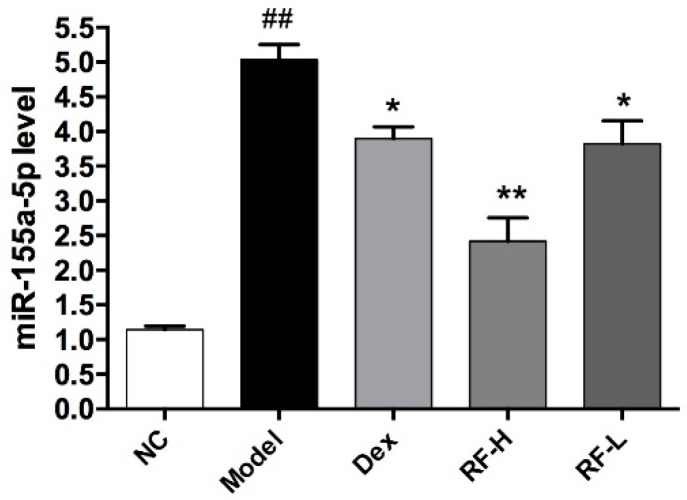
The effects of Renifolin F on the expression of microRNA 155 in the lung. Normal control, saline-induced mice only. Model, OVA-induced mice. Dex, OVA-induced mice treated with Dex (1.0 mg/kg). RF-H, OVA-induced mice treated with Renifolin F (3.0 mg/kg). RF-L, OVA-induced mice treated with Renifolin F (1.5 mg/kg). Statistics represented as the mean ± SEM (*n* = 6). ## *p* < 0.01 vs. Normal control, * *p* < 0.05 vs. Model and ** *p* < 0.01 vs. Model.

## Data Availability

The data that support the findings of this study are available from the corresponding author upon reasonable request.

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
