# Peer review of "Therapeutic Effect of Renifolin F on Airway Allergy in an Ovalbumin-Induced Asthma Mouse Model In Vivo"

_molecules, 2022, doi:10.3390/molecules27123789_

Round 1

Reviewer 1 Report

The manuscript presented by Yang and colleagues shows the effect of Renifolin F on airway allergy in a mouse 2 OVA-induced asthma. Although the biological tests applied are classic, the manuscript was well written, organized and shows scientific merit. This is important because it opens up the prospect of being tested in other biological models. However, the authors could improve the summary with the inclusion of the main results and verify how the journals are cited in the final list. Considering the points mentioned above, my opinion is that it could be accepted for publication in Molecules.

Author Response

Thank you so much for your comments, We carefully proof-read the manuscript to minimize errors. The changes we made in the manuscript were signed in red.

Here below is our description on revision according to the reviewers’ comments.

Point 1: The authors could improve the summary with the inclusion of the main results.

Response 1: We improved the summary with the inclusion of the main results in the paper, and signed in red.

Point 2: The authors could verify how the journals are cited in the final list.

Response 2: We have added the authors' name and title in the list and signed in red. But sorry that we have no idea about the doi, accepted date and republished date.

The new manuscript in the attachment.

Reviewer 2 Report

The manuscript entitled Therapeutic effect of Renifolin F on airway allergy in an 2 OVA-induced asthma mouse model. The topic of the manuscript is sound. However, there are some issues that authors must attend to prior to publication.

Manuscript ID: molecules-1760725

- Please add in start of the title In vivo (Italic).

In vivo must be written in italics. Revise in the manuscript

- OVA it is must be in title become full (ovalbumin), after that you can used abberviation.

- Lines 62-64 Revise the written idea and clarify.

- Figure 3 resolution not good.

What were the analyzing conditions for NMR?

- Orginal images and supplementary file contain the same information.

- Conclusion very short, please increase it about study finding.

Abbreviations:

- List of abbreviations should be inserted by the end of the manuscript before references.

- References need updated by 2021 and 2022 References. 

Author Response

Thank you so much for your reviewers’ comments, We carefully proof-read the manuscript to minimize errors. The changes we made in the manuscript were signed in red.

Here below is our description on revision according to the reviewers’ comments.

Point 1: Please add in start of the title In vivo (Italic). In vivo must be written in italics. Revise in the manuscript. OVA it is must be in title become full (ovalbumin), after that you can used abberviation.

Response 1: The title has been changed as: Therapeutic effect of Renifolin F on airway allergy in an ovalbumin-induced asthma mouse model in vivo. We adjusted in the manuscript and signed in red.

Point 2: Lines 62-64 Revise the written idea and clarify.

Response 2: We have revised and clarified the written idea for Line 62-64 as the manuscript marked in red.

Point 3: Figure 3 resolution not good.

Response 3: We have adjusted the picture for clarity.

Point 4: What were the analyzing conditions for NMR?

Response 4: The NMR of Renifolin F was test on 1H-NMR (400MHz, DMSO-d6) and 13C-NMR (100MHz, DMSO-d6).

Point 5: Orginal images and supplementary file contain the same information.

Response 5: In supplementary, we removed the chemical structural. We adjusted the order of the diagrams both in the manuscript and supplementary, and signed in red.

Point 6: Conclusion very short, please increase it about study finding.

Response 6: We have supplemented and replenished the results in the conclusion with our study finding, as marked in red.

Point 7: List of abbreviations should be inserted by the end of the manuscript before references.

Response 7: We listed the abbreviations at the end of the manuscript before references, and signed in red.

Point 8: References need updated by 2021 and 2022 References.

Response 8: Some references have been updated by 2021 and 2022, and signed in red.

The new manuscript was in the attachment.
